# A building-block design for enhanced visible-light switching of diarylethenes

Zhiwei Zhang [1], Wenhui Wang[1], Peipei Jin[2], Jiadan Xue[2], Lu Sun[3], Jinhai Huang[1], Junji Zhang [1] & He Tian [1]

Current development of light-responsive materials and technologies imposes an urgent demand on visible-light photoswitching on account of its mild excitation with high penetration ability and low photo-toxicity. However, complicated molecular design and laborious synthesis are often required for visible-light photoswitch, especially for diarylethenes. Worse still, a dilemma is encountered as the visible-light excitation of the diarylethene is often achieved at the expense of photochromic performances. To tackle these setbacks, we introduce a building-block design strategy to achieve all-visible-light photochromism with the triplet-sensitization mechanism. The simply designed diarylethene system is constructed by employing a sensitizer building-block with narrow singlet-triplet energy gap ($\Delta E_{ST}$) to a diarylethene building-block. A significant improvement on the photochromic efficiency is obtained as well as an enhanced photo-fatigue resistance over those under UV irradiation. The balance between the visible-light excitation and decent photochromism is thus realized, promoting a guiding principle for the visible-light photochromism.

[1] Key Laboratory for Advanced Materials and Joint International Research Laboratory of Precision Chemistry and Molecular Engineering, Feringa Nobel Prize Scientist Joint Research Center, School of Chemistry and Molecular Engineering, East China University of Science & Technology, 130 Meilong Road, Shanghai 200237, China. [2] Department of Chemistry, Zhejiang Sci-Tech University, Hangzhou 310018, China. [3] Institute of Modern Optics, Nankai University, Tianjin 300071, China. Correspondence and requests for materials should be addressed to J.Z. (email: zhangjunji@ecust.edu.cn)

Recent blooming development of smart materials with a photo-controllable manner prefers visible light as the trigger over traditional UV light, as visible light provides a mild excitation for sustainable applications and precise operation[1,2]. This requires the photo-responsive units, photoswitches, to work efficiently under visible-light irradiation. Nonetheless, one serious restriction of conventional photoswitches is that at least in one direction of the switching needs UV light, which often suffers from light-induced damaging, high scattering/low penetration and non-selective absorption/excitation through most media[1,2]. Therefore, designing photoswitches and shifting their operation wavelengths into visible-light window has become an urgent task. As a well-known P-type (thermo irreversible) photochromore, diarylethenes (DAE) possess excellent photochromic performances and photoisomer thermo-stability, qualifying as a star molecule in photoswitch family[3–7]. Apart from their classic applications in molecular data storage/memory devices[8–11], diarylethenes have recently reached to the bioscience, such as super-resolution bio-imaging[12,13] and enzyme/inhibitor mimicking[14–16], in which visible-light excitation is highly demanded. Regretfully, the development of visible-light diarylethenes still lags behind, due to the lack of effective design strategies that would guide the fabrication of diarylethenes with both visible-light responsiveness and efficient photochromism. Previous design of visible-light diarylethene mainly includes two strategies: 1. Direct excitation through HOMO-LUMO gap engineering; 2. Indirect excitation or sensitization through triplet sensitizers. The direct excitation aims at reducing the HOMO-LUMO energy gap of the open isomer and shifting the absorption band into the visible-light region, through the extension of π-conjugation on the peripheral aryl pendants or on the ethene bridge backbone[17–21]. However, elaborate molecular design is demanded and the depletion of the photochromism often occurs in most cases (e.g., significantly decreased photochromic quantum yields)[17,21–23]. Triplet-sensitized photochromism (TSP) seems to be an appealing strategy for visible-light-driven diarylethenes due to the relatively simple design, in which the conjugation extension is not necessary. One well-studied category is based on the metal-to-ligand charge-transfer (MLCT) mechanism[24–27], as a transition metal complex is introduced to the DAE core. Yet, diarylethenes with MLCT-based structures usually suffer from rather poor photocycloreversion quantum yields[20], together with the concern of high cost and potential metal contamination[28]. An alternative method is to mix diarylethenes with proper metal-free triplet sensitizers (e.g., biacetyl) to achieve visible-light TSP via a triplet-triplet energy transfer (TTET) process[5,21,29]. The main disadvantage, nonetheless, is that the TSP efficiency is rather low due to the symmetry-forbidden n-π* transition feature of commonly used biacetyl sensitizer[21]. In all, a dilemma remains unsolved in current development of visible-light-responsive diarylethenes, as a trade-off effect between the visible-light excitation and the intrinsic photochromism always occurs in all these design strategies. Inefficient photoswitching and reversibility severely affects the material properties and control precision. Besides, prolonged or intensified irradiation (even using visible light) will certainly harm the endurance of the fabricated materials.

To break these limitations and strike a balance between the structure-property relation of the visible-light photoswitching, herein we present a building-block design of triplet-sensitized photoswitching with both visible-light excitation and enhanced photochromic performances. In our design, a sensitizer building-block (9, 9-dimethyl-9, 10-dihydroacridine-2, 4, 6-triphenyl-1, 3, 5-triazine, DT, Fig. 1 and Supplementary Fig. 1) with narrow singlet-triplet energy gap ($\Delta E_{ST}$) is introduced to a conventional DAE building-block to form a diarylethene-sensitizer dyad (DAE-DT). Under the visible-light excitation ($\lambda = 420$ nm), an efficient photochromism is obtained with high photochromic quantum yields ($\Phi_{o-c}$ up to 0.40) and robust photo-fatigue resistance. The proposed intramolecular TTET mechanism is further examined and verified through transient absorption spectroscopies. Our strategy may create a visible-light DAE database for diversified design of solid-state, opto-/electronic materials and bio-systems with photo-controllable functions.

## Results

**Visible-light photochromism design strategy**. The proposed visible-light photochromism is based on the intramolecular TTET process from the triplet sensitizer to the DAE photochromore, as depicted in Fig. 1. The sensitizer presented in this work possesses a twisted donor-acceptor (D-A) structure that separates HOMO-LUMO and, consequently, leads to a narrowed gap between the singlet and triplet energy levels ($\Delta E_{ST}$, Fig. 1a)[28,30–32]. This unique feature of the sensitizer offers several advantages for visible-light photochromism: (1) The triplet population for efficient TTET process is enriched, which facilitates the intersystem crossing (ISC) and sensitization[33]. (2) The D-A structure provides a medium strong charge-transfer (CT) band in the visible-light region (~450 nm for DT, Supplementary Fig. 1, $\varepsilon = 2.0 \times 10^3$ $M^{-1}cm^{-1}$ at 420 nm) that ensures an efficient, visible-light excitation. (3) Both demands of triplet energy level matching and the visible-light excitation for the TTET-induced, visible-light photochromism are easily satisfied, benefiting from the narrow $\Delta E_{ST}$ feature of our sensitizer. Conventional sensitizers with strong visible-light absorbance usually possess much lower triplet energy levels that mismatch those of diarylethenes, resulting in the disenabling of TSP. Our strategy wards off this problem and provides a simple building-block design for the visible-light DAE photochromic system. Furthermore, our design requires no conjugation extension between the sensitizer and DAE (Fig. 1b). The electronical independent sensitizer and DAE building blocks retain their own properties and prevent the possible intramolecular electronic interactions in elongated π-conjugation[22,23], resulting in good cooperation rather than perturbation on the desired photochromic performances.

**Selection of matched DAE/sensitizer building blocks**. To select proper DAE/sensitizer building blocks with matched triplet energy levels for visible-light photochromism, the TSP of DAEs mixed with different narrow $\Delta E_{ST}$ sensitizers was first checked. The D-A type sensitizer, DT ($S_1 = 2.54$ eV, $T_1 = 2.53$ eV; Supplementary Table 1) and PT ($S_1 = 2.53$ eV, $T_1 = 2.47$ eV)[34] were mixed with a conventional DAE-1o (open isomer; $S_1 = 4.19$ eV, $T_1 = 2.49$ eV, Supplementary Table 1) at $10^{-4}$ M, respectively. As shown in Supplementary Fig. 2a, the TSP of DAE-1o/DT upon irradiation with visible light ($\lambda = 420$ nm) were detected, as a characteristic peak of the closed isomer, DAE-1c, appeared around 530 nm (identical with the closed isomer under 313 nm irradiation, Supplementary Fig. 3). Note that the DAE-1o alone is inert under 420 nm irradiation (Supplementary Fig. 3), demonstrating a possible participation of the triplet state during visible-light photochromism and the matched triplet energy levels between DAE-1o and DT. In contrast, DAE-1o/PT only exhibited slight photochromism (Supplementary Fig. 2b), indicating an unsatisfied triplet energy level matching between PT and DAE-1o. Notably, DAE-1o/DT hardly underwent photochromism in diluted solution ($2 \times 10^{-5}$ M, Supplementary Fig. 3), which presented an evident concentration dependence of photochromism in mixed system.

The triplet-sensitized photochromism mechanism was further proved by transient absorption spectroscopy, monitoring the decay of the peak at 620 nm upon nanosecond-pulsed laser excitation (Supplementary Fig. 4). The transient absorption decays of the

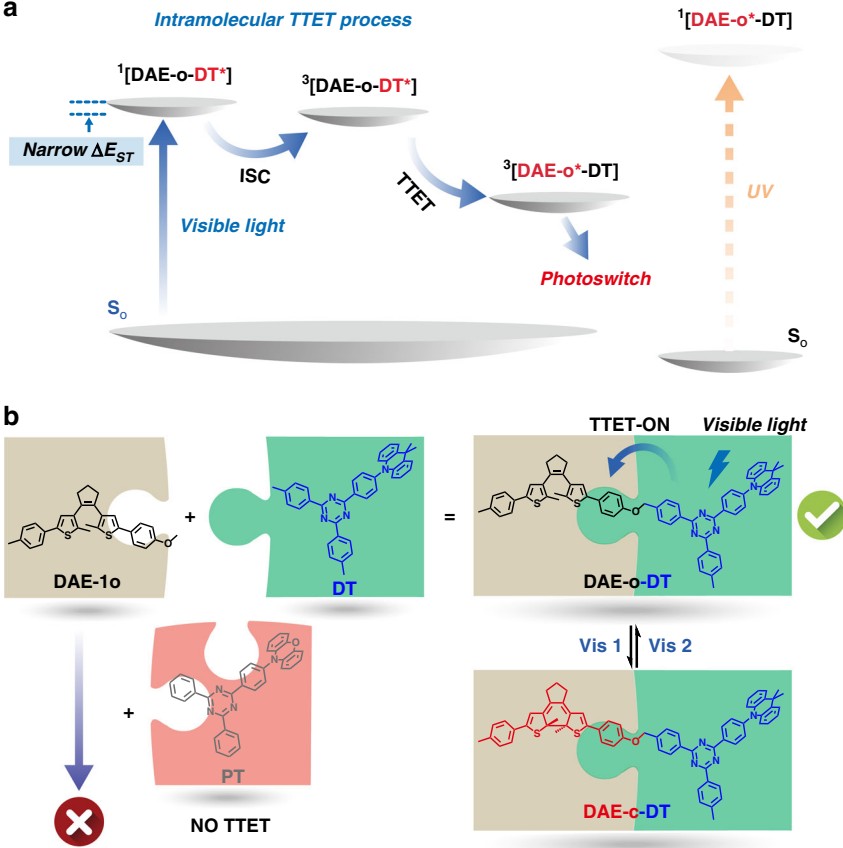

**Fig. 1** Molecular design of the visible-light photochromic DAE-o-DT and its working mechanism. **a** Illustration of the working mechanism for the intramolecular TTET-induced visible-light photochromism with narrow $\Delta E_{ST}$ sensitizer. **b** Structures of the investigated building blocks and the photoswitching with visible lights in both directions

sensitizer DT at 620 nm under both aerated and deaerated conditions were measured as well (Supplementary Fig. 5). A faster decay, with a lifetime of 65 ns, was detected in aerated toluene, as compared with 2973 ns in deaerated toluene, confirming that the transient absorption band at 620 nm could be attributed to the excited triplet state of DT. A concentration-dependent excited triplet lifetime decay was then examined (Supplementary Fig. 6a). The resulted linear Stern–Volmer plot verified the intermolecular TTET process between DT and DAE-1o (Supplementary Fig. 6b). Though a strong concentration dependence on the photochromism of DAE-1o/DT mixture was indicated, the energy level matching between DAE-1o and the sensitizer DT was confirmed to construct the DAE-DT dyad for visible-light photochromism study.

**Visible-light photochromic performances of DAE-DT.** Based on the preliminary results, DAE-DT was designed by connecting the properly selected DT sensitizer and DAE building blocks with a 1:1 ratio (Fig. 1). DAE-o-DT (open isomer of DAE-DT) exhibits a strong absorption band at around 380 nm (absorbance of DT, Fig. 2a), which extends to ca. 450 nm (CT band of DT) and thus ensures the visible-light excitation. The visible-light photochromism of DAE-DT was then investigated. As anticipated, upon visible-light irradiation at $\lambda = 420$ nm, a gradually increasing absorption band was observed around 530 nm, accompanied with color changes from pale yellow to deep red (Fig. 2b). This absorption band is identical to that formed under UV light-induced cyclization (Fig. 2a, blue dash line), indicating the generation of the closed form, DAE-c-DT (Fig. 2a, red line). The photostationary state was reached after 30 s of irradiation. The closed isomer exhibited a good thermo-stability for more than

1 week under room temperature in the dark, qualifying as a good P-type DAE molecule (Supplementary Fig. 7). Irradiation with another visible light ($\lambda > 550$ nm, 3 min) triggered the photocycloreversion (Fig. 2b). A significantly improved photo-fatigue resistance was discovered under the excitation of 420 nm light (Fig. 2c). While UV triggered photochromism showed a gradual degradation (~50%) after four cycles of alternate UV-vis irradiation, photochromism via visible light led to a much better photo-stability without obvious degradation at least for 10 cycles. The robust photo-fatigue resistance might be resulted from the mild visible light condition as well as the triplet process, which restrains the pathway to the photo-byproduct[5].

To evaluate the high efficiency of the visible-light photochromism, quantum yields and conversion ratios (Table 1 and Supplementary Tables 2–4, Supplementary Fig. 8–10) were determined. The photocyclization quantum yields with 420 nm excitation were calculated as $\Phi_{o-c} = 0.37$ (in toluene) with open-to-close conversion ratio of 80%. To our delight, the photochromic efficiency of this TSP system under visible-light excitation was higher than the UV-induced ring closure efficiency ($\Phi_{o-c} = 0.26$ with open-to-close conversion ratio of 81%), and comparable with that of the bare DAE-1o core ($\Phi_{o-c} = 0.36$ with photocyclization reversion of 85%). The photocycloreversion quantum yield $\Phi_{c-o}$ was determined as 0.006 ($\lambda = 546$ nm), almost identical to the DAE-1o core ($\Phi_{c-o} = 0.007$). The resulted photocyclization/photocycloreversion quantum yields indicate a much enhanced visible-light photochromism based on our strategy, which is usually suppressed in previous designs of visible-light DAEs[17,21–23]. Besides, compared with the mixed system ($2 \times 10^{-5}$ M, n/n = 1:1, Supplementary Fig. 3, green dash

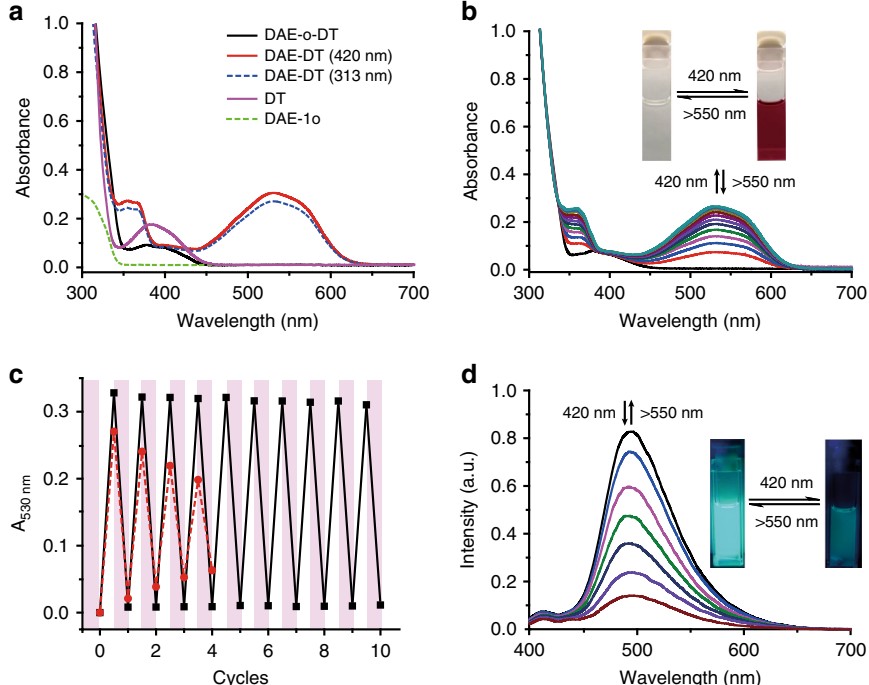

**Fig. 2** Photochromic properties of the visible-light-driven DAE-o-DT in solution. **a** Absorption spectra changes of DAE-o-DT ($2.0 \times 10^{-5}$ M) in deaerated toluene before (black) and after irradiation with 313 nm (blue dash) and with 420 nm (red solid); absorption spectra of DT ($4.0 \times 10^{-5}$ M, purple) and DAE-1o ($1.0 \times 10^{-5}$ M, green dash). **b** The photochromic performance of DAE-o-DT ($2.0 \times 10^{-5}$ M) under alternate 420 nm/>550 nm irradiation. Insets: photographs of color changes of DAE-o-DT in deaerated toluene before and after irradiation. **c** Absorbance of DAE-DT ($2.0 \times 10^{-5}$ M) in PSS at 530 nm in deaerated toluene during repetitive switching cycles consisting of alternate 420 nm/>550 nm irradiation (triplet excitation, black) and 313 nm/>550 nm irradiation (singlet excitation, red). Irradiation was carried out by an Hg/Xe lamp (200 W) equipped with a narrow band interference filter for $\lambda = 420$ nm and a broad band interference filter for $\lambda > 550$ nm. **d** The photoswitchable fluorescence performance of DAE-DT ($2.0 \times 10^{-5}$ M) under alternate 420 nm/ >550 nm irradiation. Insets: photographs of fluorescence changes of DAE-DT in deaerated toluene before and after irradiation

**Table 1 Photoreaction quantum yields of diarylethenes[a] at the depicted wavelengths ($\lambda$)[b]**

| Photoreaction | $\lambda$ (nm) | $\Phi$ |
|---|---|---|
| DAE-1o → DAE-1c | 313 | 0.36 |
| DAE-o-DT → DAE-c-DT | 313 | 0.26 |
| DAE-1o/DT[c] | 420 | 0.003 |
| DAE-o-DT → DAE-c-DT (argon bubbling) | 420 | 0.37 |
| DAE-o-DT → DAE-c-DT (oxygen bubbling) | 420 | 0.23 |
| DAE-o-DT → DAE-c-DT (anthracene) | 420 | 0.18 |
| DAE-3C-DT-o → DAE-3C-DT-c (argon bubbling) | 420 | 0.30 |
| DAE-1c → DAE-1o | 546 | 0.007 |
| DAE-c-DT → DAE-o-DT | 546 | 0.006 |

[a][DAE-1o] = $2 \times 10^{-5}$ M, [DAE-o-DT] = $2 \times 10^{-5}$ M
[b]In toluene at 300 K
[c][DAE-1o] = [DT] = $2 \times 10^{-5}$ M

line), the covalently-linked DAE-DT demonstrates a much higher efficiency of TSP under visible-light excitation. The covalent bond brings the energy donor (sensitizer) and acceptor (DAE) together and the intramolecular rather than intermolecular TTET dominates. The concentration dependence as well as the diffusion effect is eliminated, further improving the visible-light TSP.

Meanwhile, the emission peak at 495 nm of DAE-o-DT was quenched upon irradiation with 420 nm light and the PSS was almost non-fluorescent, probably due to the efficient fluorescence resonance energy transfer (FRET) between DT and DAE-1c moieties. The emission recovered to its original intensity after irradiation with >550 nm light (Fig. 2d). The photoluminescence quantum yields were further investigated to validate the FRET

process. DAE-o-DT is endowed with a photoluminescence quantum yield of $\Phi = 0.26$, while the quantum yield of resultant PSS upon irradiation with 420 nm is quenched to around $\Phi = 0.02$, confirming again the efficient intramolecular FRET in DAE-c-DT. The fluorescence-switching of DAE-DT could be conducted more than ten cycles with negligible degradation (Supplementary Fig. 11).

**Mechanism studies of the triplet-sensitized photochromism.** To validate the mechanism of the intramolecular TTET process that contributes to the visible-light photochromism, different photocyclization performances of DAE-o-DT were carried out under argon bubbling, oxygen bubbling and addition of anthracene as the triplet quencher (Fig. 3a), respectively. The photochromism of DAE-o-DT was significantly affected in oxygen bubbling and quencher containing samples, as the photocyclization quantum yields fell to $\Phi_{o-c} = 0.23$ and 0.18, respectively (Table 1). The results suggest that the triplet state plays a crucial role in the photocyclization of DAE-o-DT.

The transient absorption spectroscopy was again applied to get a deeper insight into the intramolecular TTET mechanism for DAE-DT, through monitoring the decay of the peak at 620 nm upon nanosecond-pulsed laser excitation (Supplementary Fig. 4). After argon-bubbling degassing cycles, the transient absorption decays of DAE-o-DT at 620 nm were examined (Fig. 3b). To our delight, the excited triplet state lifetime of the sensitizer moiety decreased dramatically to 293 ns for DAE-o-DT (2973 ns for DT alone), which demonstrated the quenching of the DT excited triplet state. To further prove the TTET-induced photochromism, a control compound DAE-3C-DT with a three carbon spacer between DAE acceptor and DT donor was synthesized (Supplementary Fig. 1). DAE-3C-DT exhibits similar visible-light photochromism

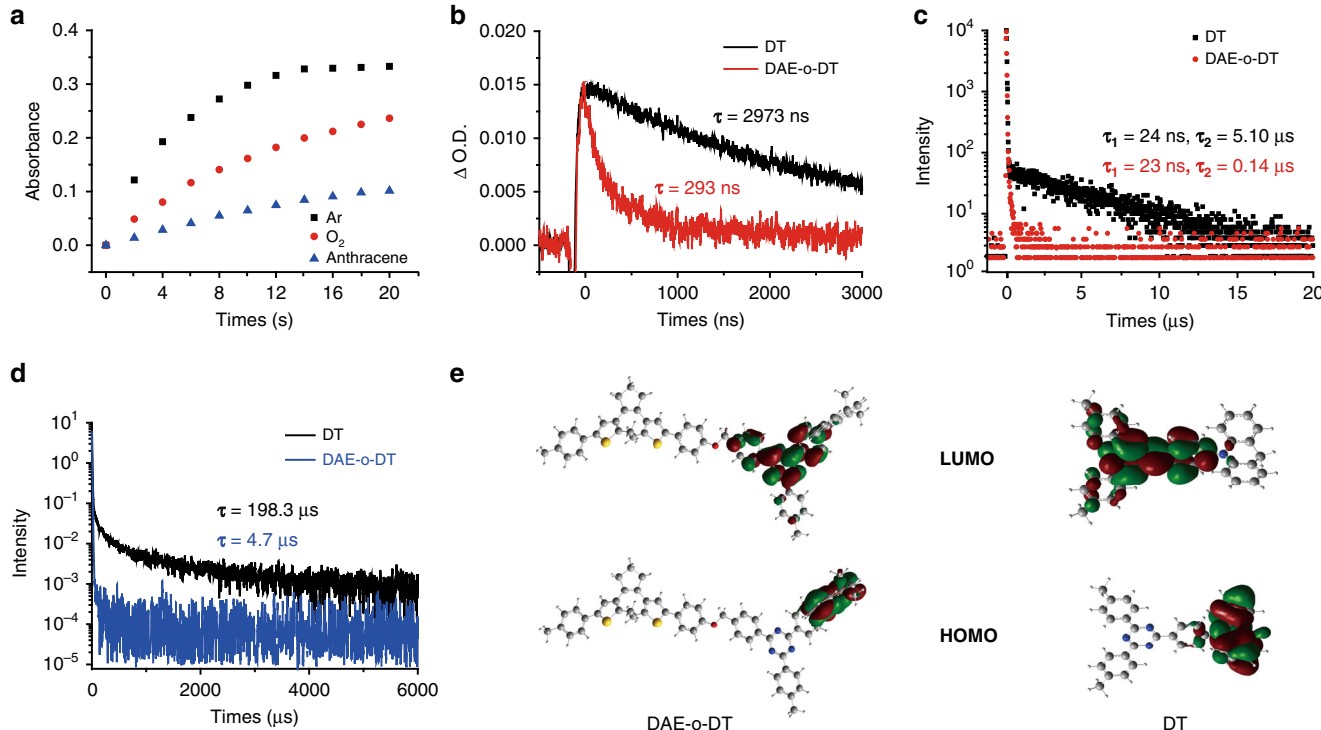

**Fig. 3** The mechanism investigation of visible-light-driven DAE-o-DT. **a** Photocyclization performances of DAE-DT ($2.0 \times 10^{-5}$ M) in argon bubbling toluene (black square), oxygen bubbling toluene (red circle) and toluene with triplet quencher anthracenes (1.0 mM, blue triangle), respectively. **b** Transient absorption decays of DT ($1.0 \times 10^{-4}$ M) and DAE-o-DT ($1.0 \times 10^{-4}$ M) at 620 nm. **c** The luminescence lifetime spectra of DT ($2.5 \times 10^{-4}$ M) and DAE-o-DT ($2.5 \times 10^{-4}$ M) at 495 nm. **d** The emission decays of DT ($2.5 \times 10^{-4}$ M) and DAE-o-DT ($2.5 \times 10^{-4}$ M) at 77 K in toluene. **e** Frontier molecular orbitals of DAE-o-DT and DT by B3LYP functional and the 6-31 G(d) basis set, respectively

(Supplementary Fig. 12a) with a slightly decreased photocyclization quantum yield of $\Phi_{o\text{-}c} = 0.30$ (Table 1). The relatively lower quantum yield indicates a less efficient TSP in the control compound, in which a three-carbon bridge is inserted between the sensitizer and DAE. The transient absorption spectroscopic measurement was then performed and a much longer lifetime of DT excited triplet state was demonstrated (673 ns, Supplementary Fig. 12b), revealing the evident distance effect of Dexter-type energy transfer for TTET[26]. Hence, the significantly shortened lifetimes and evident distance effect confirmed the existence of the intramolecular TTET between the DAE and DT moieties in DAE-DT[35–37].

The luminescence lifetime of DAE-o-DT as well as the DT sensitizer was also measured to further prove the intramolecular TTET mechanism. The narrow $\Delta E_{ST}$ feature of DT allows the reverse intersystem crossing (RISC) process from the triplet to the singlet state[30]. This phenomenon has been widely applied in designing thermally activated delayed fluorescence (TADF) materials[30,38–40]. The fluorescence of such molecule consists of two components: prompt fluorescence (~ns lifetime) and delayed fluorescence (~μs lifetime). The latter is attributed to the RISC process[30]. Accordingly, we investigated the delayed fluorescence lifetime of DAE-o-DT and DT, respectively. For DT alone, the lifetimes of prompt and delayed fluorescence were 24 ns and 5.10 μs, respectively, consistent with the typical character of TADF materials[38]. Interestingly, compared with DT, the lifetime of delayed fluorescence for DAE-o-DT was remarkably reduced to 0.14 μs, while that of prompt fluorescence did not vary too much (23 ns), demonstrating the existence of the intramolecular TTET process (Fig. 3c). Moreover, the decays of the photoluminescence lifetime at 77 K decreased dramatically from 198.3 μs for DT alone to 4.7 μs for DAE-o-DT, which further confirmed the existence of triplet energy transfer in this visible-light photochromic system (Fig. 3d).

Theoretical calculations were carried out at the B3LYP/6-31 G (d) level of theory to further study the visible-light TSP. As exhibited in Fig. 3e and Supplementary Tables 5 and 6, the HOMO and LUMO of the DAE-o-DT are mainly localized on the dihydroacridine unit and triazine unit of the DT moiety, respectively, which is similar to the DT sensitizer. This result reveals the direct excitation of the DT moiety of DAE-o-DT upon 420 nm irradiation. The above studies give a clear picture on the whole process of visible-light photochromism via intramolecular TTET mechanism (Fig. 1a). First, the sensitizer DT is excited by the visible light ($\lambda = 420$ nm) and the excited triplet state $^3$[DAE-o-DT*] is formed through ISC from the excited singlet state $^1$[DAE-o-DT*]. Upon the sequential intramolecular TTET, the energy is transferred from $^3$[DAE-o-DT*] to $^3$[DAE-o*-DT], which results in the photocyclization of DAE-o-DT and yields the closed DAE-c-DT via a triplet-sensitized pathway. The visible light could not induce the photochromism via the singlet pathway, as the DAE molecule cannot reach the excited singlet state $^1$[DAE-o*-DT] through visible-light excitation.

**Solvent effects on the triplet-sensitized photochromism.** The D-A structure of the DT sensitizer prompted us to further investigate the solvatochromic effect on the visible-light TSP. First, emission performances of DAE-o-DT were examined in different solvents with increasing polarity (Fig. 4a). An obvious solvatochromic effect on the emission spectra was observed with a redshift of the fluorescence peak from 444 nm in cyclohexane to 598 nm in acetone, indicating a lowering of excited energy levels with the increasing solvent polarity. Accordingly, a declined visible-light TSP performance was determined (Fig. 4b and Supplementary Fig. 13), as the photocyclization quantum yield dropped from $\Phi_{o\text{-}c} = 0.40$ in cyclohexane to $\Phi_{o\text{-}c} = 0.008$ in

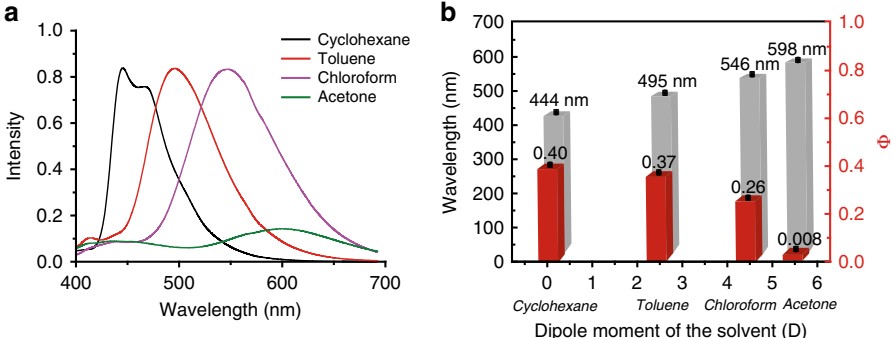

**Fig. 4** Solvent-dependent photochromic properties. **a** Fluorescence emission spectra of DAE-o-DT in different solvents. **b** Solvent-dependent photochromic performance of DAE-o-DT ($2.0 \times 10^{-5}$ M, red column) and the emission wavelength of DAE-o-DT in different solvents ($\lambda_{ex} = 365$ nm, gray column)

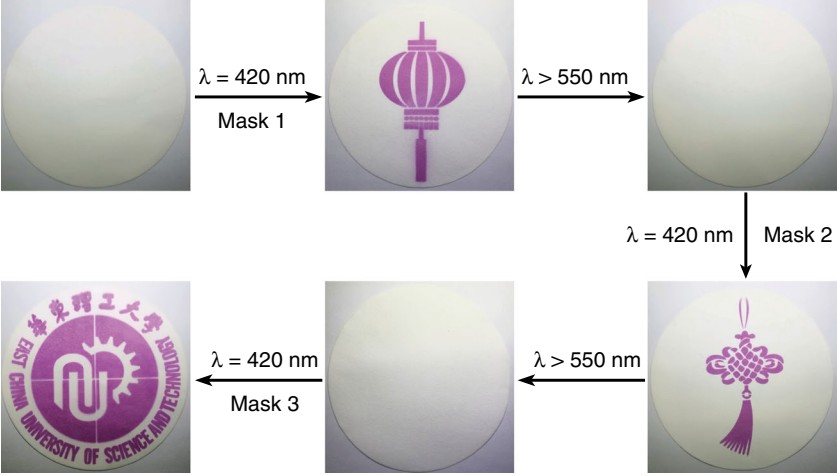

**Fig. 5** All-visible-light patterning application of DAE-DT on a filter paper. A series of images (Chinese lantern, Chinese knot and the badge of East China University of Science and Technology) were sequentially written onto and erased from the same filter paper using different masks with 420 nm irradiation for 5 min and >550 nm irradiation for 15 min, respectively

acetone (Supplementary Table 4). Though DAE-DT presents the best visible-light photochromism in cyclohexane, the relatively lower solubility may hamper its further applications. As a result, toluene was selected as the solvent in this work.

**All-visible-light photoswitchable patterning**. With this highly efficient, fatigue-resistant, visible-light photochromic DAE-DT in hand, we developed an all-visible-light patterning application for information storage medium. We prepared the filter paper by soaking it in a DAE-o-DT solution (1 mM) and drying afterward. By exposing the filter paper to the recording visible light ($\lambda = 420$ nm) locally through mask 1, specific images could be successfully patterned (Fig. 5). The image could be erased as the erasing visible light ($\lambda > 550$ nm) was imposed. Multiple images could be consecutively patterned and erased with respective masks under all-visible-light recording-erasing cycles, and no color fading was observed. Meanwhile, a vague photoswitchable patterning was obtained by soaking the filter paper with a mixed solution of DAE-1o and DT (1 mM, molar ratio = 1:1, Supplementary Fig. 14). The much attenuated photochromism of DAE-1o/DT in filter paper is probably due to the reduced mobility in solid state that significantly affects the intermolecular TTET process. The result indicates again the highly efficient intramolecular TTET of DAE-DT for all-visible-light photoswitchable patterning in solid state. These results present a promising application of DAE-DT for future light-manipulative data storage.

## Discussion

We developed an all-visible-light diarylethene system by introducing a narrow $\Delta E_{ST}$ type sensitizer to the DAE core. Taking advantages of the unique features of the sensitizer (narrow $\Delta E_{ST}$, strong visible-light absorption, etc.) and the intramolecular TTET process, the newly designed DAE-DT was endowed with a remarkable visible-light photochromism of high efficiency and robust fatigue resistance. Therefore, the long-time dilemma in previous design strategies—the trade-off effect between the visible-light excitation and photochromic performances, have been resolved. This smart strategy may fulfill the requirements for further development and modulation of visible-light photoswitches in both molecular structure designs and corresponding applications. First, the narrow $\Delta E_{ST}$ feature of the twisted D-A sensitizer overcomes the issue of triplet energy mismatch in conventional visible-light excited sensitizer (due to the wide $\Delta E_{ST}$)[34]. With the booming development of organic luminescence[41–43], numbers of newly-designed narrow $\Delta E_{ST}$ compounds are being presented every year, thus enriching our sensitizer database and offering a simple and versatile design of visible-light DAE systems. Second, the building-block strategy can facilitate the bespoke construction of DAE-sensitizer systems with demanded visible-light excitation wavelengths, by appropriately selecting sensitizer and diarylethene building blocks from the database. Third, the modular building of DAE-sensitizer system can also be directed by various means, e.g., supramolecular self-assembly[44] and polymer chemistry[45], which may inspire further

design of water-soluble, all-visible-light triggered DAEs. Under such circumstance, a wide research direction might be open to biomaterials and biotechnologies operating with visible-light photoswitches, such as super-resolution imaging[46], in which visible-light switching and water-solubility are highly demanded.

Current diarylethene system still needs to be improved as the visible light wavelength used in this work ($\lambda = 420$ nm) doesn't reach the best wavelength range for materials and biological applications (biological window: 600–1200 nm). Delicately selecting diarylethene cores with lower triplet energy levels and corresponding narrow $\Delta E_{ST}$ sensitizers is to be carried out to achieve red-light triggered photochromism that enables deeper tissue penetration during bio-imaging and modulation, which would be our next work.

## Methods

**General methods.** All starting chemicals were commercially available and analytical purity without further treatment. Solvents were distilled and dried or degassed if necessary before use. Thin-layer chromatography (TLC) analysis was performed on silica gel plates and column chromatography was conducted using silica gel column packages purchased from Yantai HuangHai Chemical (China). NMR spectra were recorded on Bruker AM-400 spectrometers with tetramethylsilane as an internal reference, CDCl$_3$ as the solvent. High resolution mass (HRMS) spectra were measured on a Waters LCT Premier XE spectrometer. For rewritable application on filter papers, a portable PLS-LED light (Beijing perfectlight technology co., LTD, 420 nm, 30 W) and a commercial flashlight (5 W) with the interference filter (>550 nm) were used and the distances between the light sources and the filter paper were 20 and 2 cm, respectively. Except where noted, all tests were carried out at room temperature.

**Preparation of DAE dyads.** DAE dyads were prepared through a two-step Williamson ether synthesis and Buchwald–Hartwig amination, with a total yield of ca. 80% (Supplementary Fig. 1). DAE-1o was synthesized as the reference photochromic core (Supplementary Fig. 1). The target molecules and all reaction intermediates were fully characterized by NMR and HRMS spectra (see Supplementary Methods for details).

**Spectroscopy measurements.** UV/Vis spectra were recorded on Varian Cary 500 (1 cm quartz cell). Fluorescence emissions were measured on Varian Cary Eclipse and the excited wavelength was 365 nm. Nanosecond transient absorption measurements were performed on LP-920 laser flash photolysis spectrometer setup (Edinburgh Instruments, UK). Excitation was performed using the third harmonic (355 nm, 100 mJ, 10 ns, 10 Hz) of a Q-switched Nd:YAG laser. The probe light was provided by a 450 W Xe arc lamp. These two light beams were focused onto a 1 cm quartz cell. The signals, which were analyzed by a symmetrical Czerny-Turner monochromator, were detected by a Hamamatsu R928 photomultiplier, and then processed via an interfaced computer and analytical software. All samples were prepared with argon bubbling and kept under an argon atmosphere. The transient photoluminescence characteristics were recorded using a Quantaurus-Tau fluorescence lifetime measurement system (C11367-03, Hamamatsu Photonics Co., Japan) with the TCC900 mode and a 371 nm LED excitation source under vacuum at room temperature. Phosphorescence emissions were carried on FLS 920 spectrofluorometer (Edinburgh Instruments, UK) in toluene at 77 K cooling by liquid nitrogen. The photochromic reaction as well as quantum yields was carried out by irradiation using an Hg/Xe lamp (Hamamatsu, LC8 Lightningcure, 200 W) equipped with a narrow band interference filter (Shenyang HB optical Technology) for $\lambda = 313$ nm, $\lambda = 420$ nm, and a broad band interference filter (Shenyang HB optical Technology) for $\lambda > 550$ nm.

**Photocyclization conversion ratio measurement.** The ratio of the equilibrium concentrations of the open form ($C_o$) and closed forms ($C_c$) at a given photostationary state (PSS) is expressed as follows:[47,48]

$$\frac{C_o}{C_c} = \frac{\varphi_{c \to o} \times \varepsilon_c}{\varphi_{o \to c} \times \varepsilon_o} = \frac{\varphi_{c \to o} \times A_c}{\varphi_{o \to c} \times A_o} \quad (1)$$

where $\varepsilon_o$ and $\varepsilon_c$ are the molar absorption coefficients of the open and closed forms, $A_o$ and $A_c$ are the absorption of a sample of same concentration containing only the open or closed form, $\Phi_{c \to o}$ and $\Phi_{o \to c}$ are quantum yields of cycloreversion and cyclization, respectively. By comparing the PSS's obtained under irradiation at two different wavelengths $\lambda_1$ and $\lambda_2$, a couple of equations of type (1) are obtained as follows (assuming that the ratio $\Phi_{c \to o}/\Phi_{o \to c}$ does not change with the irradiation wavelength):

$$\frac{C_{o1}/C_{c1}}{C_{o2}/Cc_2} = \frac{A_{c1}/A_{o1}}{A_{c2}/A_{o2}} \quad (2)$$

Equal to:

$$\frac{1 - \alpha_1/\alpha_1}{1 - \alpha_2/\alpha_2} = \frac{A_{c1}/A_{o1}}{A_{c2}/A_{o2}} \quad (3)$$

where $\alpha_1$ and $\alpha_2$ are the photocyclization conversion ratios in PSS state at the given wavelength of $\lambda_1$ and $\lambda_2$, respectively. The absorbance $A$ measured at any particular wavelength $\lambda$ of a mixture of open and closed forms was introduced, while the overall concentration $C_o + C_c$ is constant and the formula can be obtained as follows: where $\alpha_1$ and $\alpha_2$ are the photocyclization conversion ratios in PSS state at the given wavelength of $\lambda_1$ and $\lambda_2$, respectively. The absorbance $A$ measured at any particular wavelength $\lambda$ of a mixture of open and closed forms was introduced, while the overall concentration $C_o + C_c$ is constant and the formula can be obtained as follows:

$$A_c = A_o + \frac{A - A_o}{\alpha}. \quad (4)$$

Combining Eq. (3) with Eq. (4), we can obtain:

$$\frac{1 - \alpha_1/\alpha_1}{1 - \alpha_2/\alpha_2} = \frac{1 + \Delta_1/\alpha_1}{1 + \Delta_2/\alpha_2} \quad (5)$$

where $\Delta = (A - A_o)/A_o$ is the relative change of absorbance observed when a solution of open form is irradiated to the PSS. Moreover, the ratio $\rho = \alpha_1/\alpha_2$ of the conversion yields at two different PSS's, resulting from irradiation at two different wavelengths, is equal to the ratio of the $\Delta$'s measured at any given wavelength (the wavelength that maximizes the $\Delta$'s is usually chosen). Equating and developing Eq. (5) yields the final formula:

$$\alpha_2 = \frac{\Delta_1 - \Delta_2}{1 + \Delta_1 - \rho(1 + \Delta_2)} \quad (6)$$

where all the parameters $\Delta$ and $\rho$ are experimentally accessible. The numerical value determined by this equation may then be used to calculate the absorption spectrum of the pure closed form by means of Eq. (4).

**Calculation of photochromic reaction quantum yields.** The photoreaction quantum yields of DAEs at 313 nm, 420 and 546 nm were measured with the potassium ferrioxalate (K$_3$[Fe(C$_2$O$_4$)$_3$]) as the actinometer[49]. The photocyclization and photocycloreversion quantum yields were calculated based on the following equation:

$$\Phi_x = \frac{\Delta A/\Delta t}{(Nh\nu/t) \times \varepsilon_x \times F_x} \quad (7)$$

where $\Delta A/\Delta t$ is the change of absorbance upon irradiation at detective wavelength, $\varepsilon_x$ is the molar extinction coefficient at detective wavelength ($\varepsilon_{530 \text{ nm}} = 17,906$ M$^{-1}$ cm$^{-1}$ for DAE-1c, 19,000 M$^{-1}$ cm$^{-1}$ for DAE-c-DT) and F$_x$ is the mean fraction of light absorbed, the value of which is 1–10$^{-A}$. For photocycloreversion quantum yields, it should be the absolute value of $\Phi_x$.

**Photoswitchable patterning application.** 4.9 mg DAE-o-DT or the mixture of 2.28 mg DAE-1o and 2.72 mg DT was dissolved in 5 mL of toluene. The solution was purged with argon for 10 min to remove the oxygen. Then a piece commercial filter paper ($d = 11$ cm) was soaked into the solution for 5 min. Then keep the wet paper in a vacuum chamber for 30 min to remove completely residual solvents. The masks of Chinese lantern, Chinese knot and the badge of East China University of Science and Technology were all bespoke commercially.

## Data availability

The data that support the findings of this study are available from the corresponding author upon request.

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

## Acknowledgements
The project is supported by Shanghai Municipal Science and Technology Major Project (Grant No. 2018SHZDZX03) and the international cooperation program of Shanghai Science and Technology Committee (17520750100). The authors acknowledge the financial support from the NSFC (21420102004, 21878086) and the Fundamental Research Funds for the Central Universities (222201717003). Z.Z. and J.Z. thank C.D. and Q.Z. for assistance with the measurement of photoluminescence lifetimes.

## Author contributions
Z.Z. synthesized all materials, performed all photophysical property measurements in solution and analyses, and prepared the paper. W.W. assisted the synthesis of some intermediate products and the filter paper application. P.J. and J.X. assisted the nanosecond transient absorption measurements. L.S. performed the theoretical calculations. J.H. assisted the design of the synthetic route. J.Z. and H.T. designed and supervised the research and wrote the paper. All authors discussed the results and commented on the manuscript.

## Additional information

**Competing interests:** The authors declare no competing interests.

