## [Peer Review File · Nature Communications]

Reviewers' comments:

Reviewer #1 (Remarks to the Author):

The manuscript details the development of a donor-acceptor photochromic dyad where the donor is a sensitizer that allows for a fully visible light photoswitch based on triplet-triplet ET. The experiments are well realized with adequate controls to determine the mechanism and with definite demonstration of the application of the system including in a solid-support. The general interest is large and the area of research is of considerable novelty. I recommend publication in Nature Communication after some minor improvements.

1) The SI has an excellent breakdown of how you calculate the photostationary state for DAE-1o and DAE-o-DT. I believe these values should be included in the main text as they are of considerable interest to the reader. I would also like to see the DAE-o-DT at 313 just to ensure there is no relative change (though I assume there is minimal as you state that the QY was unmodified). I also would like to see these values supported by NMR structures of the DAE-c-DT (along with the other closed DAE) which are not reported in the SI.

2) Some additional references and discussion points:

a. You cite the Roubinet and Uno papers discussing the use of fluorescent DAE but there is little discussion on how applicable your sensitizer would be in such systems, please expand a bit.
b. There is some comments about a lack of solubility in cyclohexane and clearly water is an issue as there are no values reported. Beyond solubility would the TTET sensitizer function in an aqueous environment? Would degassing help or would the water itself decrease the excited state lifetime of the DT? There have been some recent works in which polymers and fibrils have been used to create photoswitchable environments for DAE even in aqueous buffers. Could your system be used within these environments?

i. In Situ Photoconversion of Multicolor Luminescence and Pure White Light Emission Based on Carbon Dot-Supported Supramolecular Assembly

Huang Wu, Yong Chen, Xianyin Dai, Peiyu Li, J. Fraser Stoddart, and Yu Liu
Journal of the American Chemical Society Just Accepted Manuscript

ii. Quantum Dots as Templates for Self-Assembly of Photoswitchable Polymers: Small, Dual-Color Nanoparticles Capable of Facile Photomodulation

Sebastián A. Díaz, Luciana Giordano, Julio C. Azcárate, Thomas M. Jovin, and Elizabeth A. Jares-Erijman

Journal of the American Chemical Society 2013 135 (8), 3208-3217

Reviewer #2 (Remarks to the Author):

This work by Z. Zhang et al. is based on the innovative idea of using TADF dye as the triplet photosensitizer for DAE photochromism. Intriguing features of intersystem crossing and triplet energy transfer are well combined to give fairly efficient visible light photochromism for both ring closing and opening reaction. The idea is novel and the synthesized DAE-TADF dyad molecule (DAE-DT) shows the highly reversible and fatigue resistant photochromism which outperforms the state-of-the-art triplet sensitized photochromism reported so far. Therefore, this work deserves publication in Nature Communication, if and only if the following issues are appropriately addressed in the revised submission.

1. Photochromism of DAE/DT mixture system is also working for TTET at higher molar concentration range as shown in Supplement Figure 2(b). Therefore, the mixture system at the solid state is expected to show the good photochromic performance, which unfortunately is not demonstrated in the current manuscript. Solid state photochromic performances of DAE-DT dyad and DAE/DT 1:1 mixture system should be compared particularly for the Figure 5 type patterning demonstration. It should be clarified if the dyad performance is better or not.

2. Basic premise of the TTET energy transfer scheme shown in Figure 1(a) is that the dyad

molecule DAE-DT is a supramolecule but not a big molecule. To be unambiguously convinced of this premise, I would recommend synthesis of longer spacer DAE-DT dyad (for example -O-(CH₂)₃- linker) to reveal the distance effect of Dexter energy transfer.

3. Triplet energy levels of DAE, DT, and also the DAE-DT dyad should all be measured and used in the discussion of the TTET mechanism instead of the DFT calculated ones.

4. While the PL lifetime of DAE-DT dyad was measured and discussed, no static PLQY of it is not reported in this work. I would suggest the PLQY measurements of DAE-o-DT and DAE-c-DT. If fluorescent, its switching behavior should also be discussed.

5. TTET is competing with rISC in DAE-DT. Actual rate constants of both processes should be calculated. The rate constant ratios should be related to the temperature effect shown in Figure 4(b).

Reviewer #3 (Remarks to the Author):

This paper reports a new design for diarylethene derivatives, which undergo photochromic reactions in both directions upon irradiation with visible light. The visible light sensitivity is one of challenging targets in molecular photoswitches. The present approach is of quite attractive and worth publishing. However, there are some drawbacks to this paper from the view point of photophysical chemistry. It is recommended to take into account the following comments.

1. p 7, line 2-16: The result of the sensitization experiment is not convincing. At 420 nm only DT has absorption. This means that photoexcited DT transfers its energy to DAE-1o and the energy transfer efficiency is dependent on the absolute concentration (mol/L) of DAE-1o (not the molar ratio). Very low concentration of DAE-1o ($\sim 10^{-5}$ - 10^{-6} mol/L) can not accept the energy from the excited DT, as shown in Supplementary Figure 6. The amount of excited DT is dependent on illumination intensity and in general it is very low.

2. p 7, line 2 from the bottom: The decrease in the lifetime of DT in aerated toluene indicates the oxygen quenching rate of $8 \times 10^8 \text{ M}^{-1}\text{s}^{-1}$. Why is the oxygen quenching so inefficient ?

3. p 9, line 1-2 from the bottom: The rates have no meaning. These values are dependent on the illumination intensity.

4. p 11, line 3 from the bottom: The cyclization quantum yield of 0.23 in the presence of oxygen seems too large. It is recommended to show the transient absorption decay at 620 nm of DAE-o-DT in deaerated and aerated solutions, which may account for the oxygen effect on the cyclization quantum yield

Review 1:

The manuscript details the development of a donor-acceptor photochromic dyad where the donor is a sensitizer that allows for a fully visible light photoswitch based on triplet-triplet ET. The experiments are well realized with adequate controls to determine the mechanism and with definite demonstration of the application of the system including in a solid-support. The general interest is large and the area of research is of considerable novelty. I recommend publication in Nature Communication after some minor improvements.

Q1: *The SI has an excellent breakdown of how you calculate the photostationary state for DAE-1o and DAE-o-DT. I believe these values should be included in the main text as they are of considerable interest to the reader. I would also like to see the DAE-o-DT at 313 just to ensure there is no relative change (though I assume there is minimal as you state that the QY was unmodified). I also would like to see these values supported by NMR structures of the DAE-c-DT (along with the other closed DAE) which are not reported in the SI.*

A1: Thanks for your suggestion. We have added the conversion ratio of all DAE compounds and moved the calculated method of photostationary state for DAEs from the Supplementary Information to the Section “Method” in the Main Text, which is shown in yellow highlight as follows:

Main Text, Page 9-10:

To evaluate the high efficiency of the visible-light photochromism, quantum yields and conversion ratios (Table 1 and Supplementary Table 2-4, Supplementary Fig. 8-10) were determined. The photocyclization quantum yields with 420 nm excitation were calculated as $\Phi_{o-c} = 0.37$ (in toluene) with open-to-close conversion ratio of 80%. To our delight, the photochromic efficiency of this TSP system under visible-light excitation was higher than the UV-induced ring closure efficiency ($\Phi_{o-c} = 0.26$ with open-to-close conversion ratio of 81%), and comparable with that of the bare **DAE-1o** core ($\Phi_{o-c} = 0.36$ with photocyclization reversion of 85%).

Main Text, Method:

Photocyclization conversion ratio measurement. The ratio of the equilibrium

concentrations of the open form (C_o) and closed forms (C_c) at a given photostationary state (PSS) is expressed as follows⁴⁷⁻⁴⁸:

$$\frac{C_o}{C_c} = \frac{\varphi_{c \rightarrow o} \times \varepsilon_c}{\varphi_{o \rightarrow c} \times \varepsilon_o} = \frac{\varphi_{c \rightarrow o} \times A_c}{\varphi_{o \rightarrow c} \times A_o} \quad \text{Eq. 1}$$

where ε_o and ε_c are the molar absorption coefficients of the open and closed forms, A_o and A_c are the absorption of a sample of same concentration containing only the open or closed form, $\Phi_{c \rightarrow o}$ and $\Phi_{o \rightarrow c}$ are quantum yields of cycloreversion and cyclization, respectively. By comparing the PSS's obtained under irradiation at two different wavelengths λ_1 and λ_2 , a couple of equations of type (Eq.1) are obtained as follows (assuming that the ratio $\Phi_{c \rightarrow o} / \Phi_{o \rightarrow c}$ does not change with the irradiation wavelength):

$$\frac{C_{o1}/C_{c1}}{C_{o2}/C_{c2}} = \frac{A_{c1}/A_{o1}}{A_{c2}/A_{o2}} \quad \text{Eq. 2}$$

Equal to:

$$\frac{1 - \alpha_1 / \alpha_1}{1 - \alpha_2 / \alpha_2} = \frac{A_{c1}/A_{o1}}{A_{c2}/A_{o2}} \quad \text{Eq. 3}$$

Where α_1 , α_2 are the photocyclization conversion ratios in PSS state at the given wavelength of λ_1 , λ_2 , respectively. The absorbance A measured at any particular wavelength λ of a mixture of open and closed forms was introduced, while the overall concentration $C_o + C_c$ is constant and the formula can be obtained as follows: Where α_1 , α_2 are the photocyclization conversion ratios in PSS state at the given wavelength of λ_1 , λ_2 , respectively. The absorbance A measured at any particular wavelength λ of a mixture of open and closed forms was introduced, while the overall concentration $C_o + C_c$ is constant and the formula can be obtained as follows:

$$A_c = A_o + \frac{A - A_o}{\alpha} \quad \text{Eq. 4}$$

Combining Eq. (3) with (4), we can obtain:

$$\frac{1-\alpha_1/\alpha_1}{1-\alpha_2/\alpha_2} = \frac{1+\Delta_1/\alpha_1}{1+\Delta_2/\alpha_2} \quad \text{Eq. 5}$$

where $\Delta = (A-A_0)/A_0$ is the relative change of absorbance observed when a solution of open form is irradiated to the PSS. Moreover, the ratio $\rho = \alpha_1/\alpha_2$ of the conversion yields at two different PSS's, resulting from irradiation at two different wavelengths, is equal to the ratio of the Δ 's measured at any given wavelength (the wavelength that maximizes the Δ 's is usually chosen). Equating and developing Eq. (5) yields the final formula:

$$\alpha_2 = \frac{\Delta_1 - \Delta_2}{1 + \Delta_1 - \rho(1 + \Delta_2)} \quad \text{Eq. 6}$$

where all the parameters Δ and ρ are experimentally accessible. The numerical value determined by this equation may then be used to calculate the absorption spectrum of the pure closed form by means of equation Eq. (4).

Also, we investigated the photocyclization conversion ratios of **DAE-1o**, **DAE-o-DT** in different conditions by NMR spectra (See Supplementary Fig. 8-10). Expectedly, the obtained values are in line with those shown in the Main Text (Supplementary Table 2-3).

Supplementary Table 2. Computational photocyclization conversion ratios of the given switch isomers at the depicted wavelengths (λ_{irr}) in degassed toluene. **[DAE-1o] = [DAE-o-DT] = 2×10^{-5} M.**

Photoreaction	λ_{irr} (nm)	Conversion ratios (PSS)
DAE-1o→DAE-1c	313	85%
DAE-o-DT→DAE-c-DT	313	81%
DAE-o-DT→DAE-c-DT	420	80%

Supplementary Table 3. Experimental photocyclization conversion ratios of the given switch isomers at the depicted wavelengths (λ_{irr}) in degassed toluene by ^1H NMR.

Photoreaction	λ_{irr} (nm)	Conversion ratios (PSS)
DAE-1o→DAE-1c	313	80%
DAE-o-DT→DAE-c-DT	313	85%
DAE-o-DT→DAE-c-DT	420	77%

Supplementary Figure 8. ^1H NMR spectra of DAE-1o (3.1×10^{-3} M) (a) before and (b) after photoirradiation at 313 nm in $[\text{D}_6]\text{benzene}$.

Supplementary Figure 9. ^1H NMR spectra of **DAE-o-DT** (3.1×10^{-3} M) (a) before and (b) after photoirradiation at 313 nm in $[\text{D}_6]\text{benzene}$.

Supplementary Figure 10. ^1H NMR spectra of DAE-o-DT (3.1×10^{-3} M) (a) before and (b) after photoirradiation at 420 nm in deaerated $[\text{D}_6]$ benzene.

Q2: Some additional references and discussion points:

a. You cite the Roubinet and Uno papers discussing the use of fluorescent DAE but there is little discussion on how applicable your sensitizer would be in such systems, please expand a bit.

b. There is some comments about a lack of solubility in cyclohexane and clearly water is an issue as there are no values reported. Beyond solubility would the TTET sensitizer function in an aqueous environment? Would degassing help or would the water itself decrease the excited state lifetime of the DT? There have been some recent works in which polymers and fibrils have been used to create photoswitchable environments for DAE even in aqueous buffers. Could your system be used within these environments?

i. *In Situ Photoconversion of Multicolor Luminescence and Pure White Light Emission Based on Carbon Dot-Supported Supramolecular Assembly*

Huang Wu, Yong Chen, Xianyin Dai, Peiyu Li, J. Fraser Stoddart, and Yu Liu. *Journal of the American Chemical Society* Just Accepted Manuscript

ii. *Quantum Dots as Templates for Self-Assembly of Photoswitchable Polymers: Small, Dual-Color Nanoparticles Capable of Facile Photomodulation*

Sebastian A. D'áz, Luciana Giordano, Julio C. Azcarate, Thomas M. Jovin, and Elizabeth A. Jares-Erijman.

Journal of the American Chemical Society 2013 135 (8), 3208-3217

A2: Much appreciate your precious suggestions about improving our system in a water environment. Indeed, there have been several excellent works where efficient photochromism could be performed in water by using polymers or fibrils alternatively. Considering the TTET mechanism and the sensitivity of triplet lifetime in different polar solvents, the latter strategy is more likely to realize the visible light-driven photochromism in an aqueous environment because of the possible separation between **DAE-o-DT** and the surrounding water molecule. Inspired by these related articles (*J. Am. Chem. Soc.* 2019, 141, 6583; *Angew. Chem. Int. Ed.* 2018, 57, 3626), we created an aqueous environment for DAEs by introducing micelles made from **mPEG-DSPE** (see details below). Intriguingly, upon irradiation with 420 nm light, photochromism was successfully performed (as shown in Fig. R1). Although the photoswitching efficiency was not that perfect as shown in toluene, it can still open a window for visible-light photoswitches in aqueous solution, which is essential for their potential applications in biotechnology. Optimization of visible light photochromism in aqueous environment would be our focus in the next work. Many thanks for your constructive suggestions again!

Fig. R1 The absorption of **DAE-o-DT** loaded by **mPEG-DSPE** in an aqueous environment upon

irradiation with 420 nm light.

Preparation of the **mPEG-DSPE** micelles with **DAE-DT**:

20 mg **mPEG-DSPE** was dissolved in 2 mL deionized water completely by sonication for 20 min. Then 2 mg **DAE-DT** in 200 μ L was added to mentioned **mPEG-DSPE** solution with stirring vigorously. After sonication for another 20 min, transfer the mixture solution to dialysis bag (1000-MW cut-off) and dialyze in 0.5 L deionized water for 12 h and change the deionized water every 4 hours.

Moreover, we discussed the photoswitchable fluorescence in our **DAE-DT** system as the DT sensitizer is emissive. The photoswitchable emission of our system promotes itself as a potential candidate for super-resolution imaging. Reference 44-46 were added for introducing water-soluble DAE systems and DAE for super-resolution imaging. The expanded discussion is added as follows:

Main Text, Page 10:

Meanwhile, the emission peak at 495 nm of **DAE-o-DT** was quenched upon irradiation with 420 nm light and the PSS was almost non-fluorescent, probably due to the efficient fluorescence resonance energy transfer (FRET) between **DT** and **DAE-1c** moieties. The emission recovered to its original intensity after irradiation with > 550 nm light (Fig. 2d). The photoluminescence quantum yields were further investigated to validate the FRET process. **DAE-o-DT** is endowed with a photoluminescence quantum yield of $\Phi = 0.26$, while the quantum yield of resultant PSS upon irradiation with 420 nm is quenched to around $\Phi = 0.02$, confirming again the efficient intramolecular FRET in **DAE-c-DT**. The fluorescence-switching of **DAE-DT** could be conducted more than 10 cycles with negligible degradation (Supplementary Fig. 11).

Main Text, Discussion:

... Third, the modular building of DAE-sensitizer system can also be directed by various means, e.g. supramolecular self-assembly⁴⁴ and polymer chemistry⁴⁵, which may inspire

further design of water-soluble, all-visible-light triggered DAEs. Under such circumstance, a wide research direction might be open to biomaterials and bio-technologies operating with visible-light photoswitches, such as super-resolution imaging⁴⁶, in which visible-light switching and water-solubility are highly demanded.

Current diarylethene system still needs to be improved as the visible light wavelength used in this work ($\lambda = 420$ nm) doesn't reach the best wavelength range for materials and biological applications (biological window: 600-1200 nm). Delicately selecting diarylethene cores with lower triplet energy levels and corresponding narrow ΔE_{ST} sensitizers is to be carried out to achieve red-light triggered photochromism that enables deeper tissue penetration during bio-imaging and modulation, which would be our next work.

Main Text, Reference:

44. Wu, H. et. al. In Situ Photoconversion of Multicolor Luminescence and Pure White Light Emission Based on Carbon Dot-Supported Supramolecular Assembly. *J. Am. Chem. Soc.*, **141**, 6583-6591 (2019).

45. Díaz, S. A. et. al. Quantum dots as templates for self-assembly of photoswitchable polymers: small, dual-color nanoparticles capable of facile photomodulation. *J. Am. Chem. Soc.*, **135**, 3208-3217 (2013).

46. Park, S. J. et al. Dual stem cell therapy synergistically improves cardiac function and vascular regeneration following myocardial infarction. *Nat. Commun.*, **10**, 3123-3134 (2019).

Review 2:

This work by Z. Zhang et al. is based on the innovative idea of using TADF dye as the triplet photosensitizer for DAE photochromism. Intriguing features of intersystem crossing and triplet energy transfer are well combined to give fairly efficient visible light photochromism for both ring closing and opening reaction. The idea is novel and the synthesized DAE-TADF dyad molecule (DAE-DT) shows the highly reversible and fatigue resistant photochromism which outperforms the state-of-the-art triplet sensitized photochromism reported so far. Therefore, this work deserves publication in Nature Communication, if and only if the following issues are appropriately addressed in the revised submission.

Q1: Photochromism of DAE/DT mixture system is also working for TTET at higher molar concentration range as shown in Supplement Figure 2(b). Therefore, the mixture system at the solid state is expected to show the good photochromic performance, which unfortunately is not demonstrated in the current manuscript. Solid state photochromic performances of DAE-DT dyad and DAE/DT 1:1 mixture system should be compared particularly for the Figure 5 type patterning demonstration. It should be clarified if the dyad performance is better or not.

A1: Thanks for this point. The photochromic properties of the mixture system in solid state has been investigated. As shown in Supplementary Fig. 14, a much attenuated photochromism was obtained, which is probably due to the reduced mobility in solid state that significantly affects the intermolecular TTET process.

Supplementary Figure 14. A series of images (Chinese lantern, Chinese knot and the badge of East China University of Science and Technology) were sequentially written onto and

erased from the same filter paper with **DAE/DT** using different masks with 420 nm irradiation for 7 min and > 550 nm irradiation for 10 min, respectively.

Main Text, Page 16:

...Meanwhile, a vague photoswitchable patterning was obtained by soaking the filter paper with a mixed solution of **DAE-1o** and **DT** (1 mM, molar ratio = 1:1, Supplementary Fig. 14).

The much attenuated photochromism of **DAE-1o/DT** in filter paper is probably due to the reduced mobility in solid state that significantly affects the intermolecular TTET process. The result indicates again the highly efficient intramolecular TTET of **DAE-DT** for all-visible-light photoswitchable patterning in solid state.

Main Text, Method:

Photoswitchable Patterning Application. 4.9 mg **DAE-o-DT** or the mixture of 2.28 mg **DAE-1o** and 2.72 mg **DT** was dissolved in 5 mL of toluene. The solution was purged with argon for 10 minutes to remove the oxygen. Then a piece commercial filter paper (d = 11 cm) was soaked into the solution for 5 minutes. Then keep the wet paper in a vacuum chamber for 30 minutes to remove completely residual solvents.

Q2: *Basic premise of the TTET energy transfer scheme shown in Figure 1(a) is that the dyad molecule DAE-DT is a supramolecule but not a big molecule. To be unambiguously convinced of this premise, I would recommend synthesis of longer spacer DAE-DT dyad (for example -O-(CH₂)₃- linker) to reveal the distance effect of Dexter energy transfer.*

A2: Thanks for your suggestion. We have synthesized a control dyad **DAE-3C-DT** with three carbon linker -O-(CH₂)₃-O-. The synthetic routine is shown in Supplementary Figure 1. Please see the details in Supplementary Information.

The visible-light photochromism of **DAE-3C-DT** was firstly checked, which exhibited a similar performance as **DAE-DT** (Supplementary Figure 12a). However, a relatively decreased photocyclization quantum yield upon 420 nm irradiation was obtained ($\Phi_{o-c} = 0.30$, revised Table 1). This may be considered as the first evidence that a less efficient TTET is expected in

DAE-3C-DT.

Secondly, the transient absorption measurements were carried out for **DAE-3C-DT**. The triplet lifetimes of **DAE-DT** and **DAE-3C-DT** at 620 nm reduce remarkably from 2973 ns (**DT**) to 293 ns and 673 ns (Supplementary Figure 12b), respectively, resulted from the intramolecular TTET process. **DAE-3C-DT** endows a longer lifetime than **DAE-DT** upon introducing a longer linker $-O-(CH_2)_3-O-$, suggesting a less efficient TTET process. Corresponding to previous researches (*J. Am. Chem. Soc.*, **2008**, *130*, 7286; *J. Am. Chem. Soc.*, **1999**, *121*, 9626), the distance-dependant quenching of triplet lifetime offers strong evidence for our premise of intramolecular TTET mechanism.

The revised part in Main Text is as follows:

Main Text, Page 12:

... To further prove the TTET-induced photochromism, a control compound **DAE-3C-DT** with a three carbon spacer between DAE acceptor and DT donor was synthesized (Supplementary Fig. 1). **DAE-3C-DT** exhibits similar visible-light photochromism (Supplementary Fig. 12a) with a slightly decreased photocyclization quantum yield of $\Phi_{o-c} = 0.30$ (Table 1). The relatively lower quantum yield indicates a less efficient TSP in the control compound, in which a three-carbon bridge is inserted between the sensitizer and DAE. The transient absorption spectroscopic measurement was then performed and a much longer lifetime of **DT** excited triplet state was demonstrated (673 ns, Supplementary Fig. 12b), revealing the evident distance effect of Dexter-type energy transfer for TTET²⁶. Hence, the significantly shortened lifetimes and evident distance effect confirmed the existence of the intramolecular TTET between the DAE and DT moieties in **DAE-DT**³⁵⁻³⁷.

Supplementary Figure 12. a) The photochromic performance of **DAE-3C-DT** ($2.0 \times 10^{-5} \text{ M}$) under irradiation at 420 nm at different times; b) Transient absorption decays of **DAE-3C-DT** ($1.0 \times 10^{-4} \text{ M}$) at 620 nm in deaerated toluene.

Q3: Triplet energy levels of DAE, DT, and also the DAE-DT dyad should all be measured and used in the discussion of the TTET mechanism instead of the DFT calculated ones.

A3: Thank you very much for your advice. We have tried to obtain the triplet energy levels of **DAE**, **DT** and **DAE-o-DT** by checking the phosphorescence at 77 K in toluene. Unfortunately, only **DT** showed a considerable luminescence peak at 463 nm at 77 K, assigned to triplet energy of 2.68 eV. (see Fig. R2 below). No phosphorescence emission was obtained from **DAE** alone. Therefore, the triplet energy level of **DAE** cannot be directly measured. For the dyad **DAE-o-DT**, the weak intensity of luminescence resulted from the efficient quenching of DAE moiety.

Indeed, the measured values may be more valid than the calculated ones. Regretfully, it is not fully accessible in our system. Therefore, we still employed the calculated values for both **DAE-1o** and **DT**. Thanks again for your advice.

Fig. R2 Luminescence of **DAE-1o**, **DT** and **DAE-o-DT** at 77 K in toluene.

Q4: While the PL lifetime of DAE-DT dyad was measured and discussed, no **static PLQY** of it is not reported in this work. I would suggest the PLQY measurements of DAE-o-DT and DAE-c-DT. If fluorescent, its switching behavior should also be discussed.

A4: Many thanks for your constructive suggestion. We have measured the visible-light photoswitchable emission performances of our DAE-DT system. As expected, a good ON/OFF fluorescence switching with high fatigue-resistance (over 10 cycles) was obtained upon alternate irradiation of 420 nm/> 550 nm light (see Main Text Figure 2d and Supplementary Figure 11). The fluorescence quantum yields before and after 420 nm irradiation were calculated as $\Phi = 0.26$ and 0.02, respectively.

Figure 2 | Photochromic properties of the visible light-driven DAE-o-DT in solution. a)

Absorption spectra changes of DAE-o-DT (2.0×10^{-5} M) in deaerated toluene before (black)

and after irradiation at 313 nm (blue dash) and at 420 nm (red solid); Absorption spectra of

DT (4.0×10^{-5} M, purple) and DAE-1o (1.0×10^{-5} M, green dash); b) The photochromic

performance of DAE-o-DT (2.0×10^{-5} M) under alternate 420 nm/> 550 nm irradiations.

Insets: Photographs of color changes of DAE-o-DT in deaerated toluene before and after

irradiation; c) Absorbance of DAE-DT (2.0×10^{-5} M) in PSS at 530 nm in deaerated toluene

during repetitive switching cycles consisting of alternate 420 nm/> 550 nm irradiations (triplet excitation, black) and 313 nm/> 550 nm irradiations (singlet excitation, red). Irradiation was carried out by an Hg/Xe lamp equipped with a narrow band interference filter for $\lambda = 420$ nm and a broad band interference filter for $\lambda > 550$ nm; d) The fluorescent photoswitching performance of **DAE-DT** (2.0×10^{-5} M) under alternate 420 nm/> 550 nm irradiations. Insets: Photographs of fluorescence changes of **DAE-DT** in deaerated toluene before and after irradiation.

Supplementary Figure 11. Emission of **DAE-DT** (2.0×10^{-5} M) at 495 nm in deaerated toluene during repetitive switching cycles consisting of alternate 420 nm/> 550 nm irradiations.

The revised parts in the Main Text and Supplementary Information are as follows:

Main Text, Page 10:

Meanwhile, the emission peak at 495 nm of **DAE-o-DT** was quenched upon irradiation with 420 nm light and the PSS was almost non-fluorescent, probably due to the efficient fluorescence resonance energy transfer (FRET) between **DT** and **DAE-1c** moieties. The emission recovered to its original intensity after irradiation with > 550 nm light (Fig. 2d). The photoluminescence quantum yields were further investigated to validate the FRET process.

DAE-o-DT is endowed with a photoluminescence quantum yield of $\Phi = 0.26$, while the quantum yield of resultant PSS upon irradiation with 420 nm is quenched to around $\Phi = 0.02$, confirming again the efficient intramolecular FRET in **DAE-c-DT**. The fluorescence-switching of **DAE-DT** could be conducted more than 10 cycles with negligible degradation (Supplementary Fig. 11).

Q5: *TTET is competing with rISC in DAE-DT. Actual rate constants of both processes should be calculated. The rate constant ratios should be related to the temperature effect shown in Figure 4(b).*

A5: Thanks a lot for your suggestion. Firstly, we tried to calculate the rate constant of rISC of **DT** and **DAE-o-DT** according to the formula reported by Adachi (*Nature Photon.*, **2012**, 6, 253; *Nature Photon.* , **2014**, 8, 326.) as follows:

$$k_{rISC} = \frac{k_p \times k_d}{k_{ISC}} \times \frac{\varphi_d}{\varphi_p} \quad \text{Eq.R1}$$

where k_p and k_d are the rate constants of the prompt and delayed fluorescence components, respectively, k_{ISC} is the ISC rate constant from singlet to triplet states, and Φ_p and Φ_d are the photoluminescence quantum efficiencies of the prompt and delayed components.

So, we can calculate the values of k_p , k_d and k_{ISC} by the following formulas:

$$k_p = \frac{\varphi_p}{\tau_p} \quad \text{Eq.R2}$$

$$k_d = \frac{\varphi_d}{\tau_d} \quad \text{Eq.R3}$$

$$\varphi_p = \frac{k_p}{k_p + k_{ISC}} \quad \text{Eq.R4}$$

Where, the prompt and delayed quantum yields are calculated according to the integrated intensity of the prompt and delayed components (*Nature Photon.* , **2014**, 8, 326.). Herein, the values of Φ_p and Φ_d are 0.39, 0.32 for **DT** and 0.25 and 0.01 for **DAE-DT**, respectively. The values of k_{rISC} are obtained according to the mentioned formulas (see Table R1 below).

Table R1. The values of k_{rISC} for the investigated compounds at 300 K in deaerated toluene.

Compounds	$k_{\text{rISC}} / \text{s}^{-1}$
DT	3.2×10^4
DAE-o-DT	1.1×10^3

Compared to the bald **DT**, the dyad **DAE-o-DT** has a much lower k_{rISC} because of the competitive TTET process. We also evaluated k_{rISC} of **DAE-o-DT** at different temperature. There is no noticeable change for the luminescence delayed spectra neither in the solution at 350 K nor in the PMMA film at lower temperatures (see Fig. R3-4), which showed that negligible change of k_{rISC} occurred from 100 K to 350 K.

Fig. R3 The luminescence decays of **DAE-o-DT** (1.0×10^{-4} M) at 495 nm in deaerated toluene at 300 K and 350 K, respectively.

Fig. R4 The luminescence decays of **DAE-o-DT** in PMMA film (5 wt%) at 463 nm at different temperatures.

The temperature-independent performances made us rethink about the temperature-dependent photochromism as we claimed in the context. We figured out that the lower photocyclization quantum yields in high temperature might be attributed to the decreased thermo-stability of **DAE-c-DT**. We then checked thermal-stability of **DAE-c-DT** under 350 K and found a thermally induced cycloreversion after 1 day. Furthermore, the photochromism of **DAE-1c** under 350 K was investigated and an obviously decreased $\Phi_{o-c} = 0.27$ was obtained compared to that of $\Phi_{o-c} = 0.36$ under 300 K (see Fig. R5 and Table R2 below). Hence, in order not to confuse the readers, we would like to remove Fig. 4d and the related statements from the Main Text.

Fig. R5 The absorption decay of PSS for **DAE-DT** (1.0×10^{-4} M) monitored at 530 nm at 350 K in the dark in deaerated toluene.

Table R2. The photocyclization quantum yield of **DAE-1o** in toluene at 300 K and 350 K.

Temperature ($^{\circ}\text{C}$)	ϕ
300 K	0.36
350 K	0.27

The revised parts are as follows:

Main Text, Page 15:

Solvent Effects on the Triplet-Sensitized Photochromism. The D-A structure of the **DT** sensitizer prompted us to further investigate the solvatochromic effect on the visible-light

TSP. First, emission performances of **DAE-o-DT** were examined in different solvents with increasing polarity (Fig. 4a). An obvious solvatochromic effect on the emission spectra was observed with a red-shift of the fluorescence peak from 444 nm in cyclohexane to 598 nm in acetone, indicating a lowering of excited energy levels with the increasing solvent polarity. Accordingly, a declined visible-light TSP performance was determined (Fig. 4b and Supplementary Fig. 13), as the photocyclization quantum yield dropped from $\Phi_{o-c} = 0.40$ in cyclohexane to $\Phi_{o-c} = 0.008$ in acetone (Supplementary Table 4). Though **DAE-DT** presents the best visible-light photochromism in cyclohexane, the relatively lower solubility may hamper its further applications. As a result, toluene was selected as the solvent in this work.

Figure 4 | Solvent-dependent photochromic properties. a) Fluorescence emission spectra of **DAE-o-DT** in different solvents. b) Solvent-dependent photochromic performance of **DAE-o-DT** (2.0×10^{-5} M, red column) and the emission wavelength of **DAE-o-DT** in different solvents ($\lambda_{ex} = 365$ nm, gray column).

Supplementary Figure 13. The absorbance spectra of **DAE-o-DT** (2.0×10^{-5} M) upon irradiation at 420 nm in deaerated solvents of a) cyclohexane; b) chloroform; c) acetone.

Secondly, we tried to measure the value of TTET rate constant by nanosecond transient absorption. The most commonly used strategy is to monitor the triplet state absorption change of acceptor and calculate the rate constant of TTET according to the lifetime (*J. Am. Chem. Soc.*, **2008**, *130*, 7286; *Chem. Eur. J.* **2006**, *12*, 5840; *Inorg. Chem.* **2004**, *43*, 2779; *J. Phys. Chem. A* **2018**, *122*, 6673). Regrettably, we failed in tracing the transient absorption peak of the triplet excited state of **DAE-1o** (usually within a sub-nanosecond timescale) by our nano-transient absorption equipment, probably due to the lack of demanded resolution.

Generally, the intramolecular TTET process is very fast with a rate constant of $10^8 - 10^9 \text{ s}^{-1}$ (*J. Am. Chem. Soc.*, **2008**, *130*, 7286; *Chem. Eur. J.* **2006**, *12*, 5840; *Science* **2016**, *351*, 369), which is usually several orders of magnitudes more substantial than that of rISC ($10^3 - 10^6 \text{ s}^{-1}$) (*J. Am. Chem. Soc.* **2017**, *139*, 4042). We consider the intramolecular TTET process dominates and the rISC process is negligible in **DAE-o-DT** as a highly quenched delayed luminescence lifetime is obtained together with no apparent value changes from 100 K to 300 K.

Reviewer 3:

This paper reports a new design for diarylethene derivatives, which undergo photochromic reactions in both directions upon irradiation with visible light. The visible light sensitivity is one of challenging targets in molecular photoswitches. The present approach is of quite attractive and worth publishing. However, there are some drawbacks to this paper from the view point of photophysical chemistry. It is recommended to take into account the following comments.

Q1: *p 7, line 2-16: The result of the sensitization experiment is not convincing. At 420 nm only DT has absorption. This means that photoexcited DT transfers its energy to DAE-1o and the energy transfer efficiency is dependent on the absolute concentration (mol/L) of DAE-1o (not the molar ratio). Very low concentration of DAE-1o (~10⁻⁵ - 10⁻⁶ mol/L) can not accept the energy from the excited DT, as shown in Supplementary Figure 6. The amount of excited DT is dependent on illumination intensity and in general it is very low.*

A1: Thank you very much for your point to help us improve our manuscript. Basically, we investigate the photochromic efficiency of **DAEs** at a given concentration (~10⁻⁵ M) with different concentrations of triplet sensitizers. As the concentration of **DT** increases, it is more accessible for energy transfer from **DT** donor to **DAE** acceptor, thus inducing the photochromism and exhibiting corresponding absorption spectra. In the revised version, we investigated the photochromic properties of **DAE-1o/DT** in both high (10⁻⁴ M) and low (2×10⁻⁵ M) concentrations with either **DT** as triplet sensitizer in equivalent concentration. The apparent concentration-dependent visible-light photochromism was concluded in mixed systems. In order not to mislead the readers and make our description more convincing, we rewrote the section of “**Selection of Matched DAE/Sensitizer Building Blocks**” as follows:

Selection of Matched DAE/Sensitizer Building Blocks. To select proper DAE/sensitizer building blocks with matched triplet energy levels for visible-light photochromism, the TSP of DAEs mixed with different narrow ΔE_{ST} sensitizers were first checked. The D-A type sensitizer, **DT** ($S_1 = 2.54$ eV, $T_1 = 2.53$ eV; Supplementary Table 1) and **PT** ($S_1 = 2.53$ eV, $T_1 = 2.47$ eV)³⁴ were mixed with a conventional **DAE-1o** (open isomer; $S_1 = 4.19$ eV, $T_1 = 2.49$ eV, Supplementary Table 1) at 10⁻⁴ M, respectively. As shown in Supplementary Fig. 2a, the TSP of **DAE-1o/DT** upon irradiation with visible light ($\lambda = 420$ nm) were detected, as a

characteristic peak of the closed isomer, **DAE-1c**, appeared around 530 nm (identical with the closed isomer under 313 nm irradiation, Supplementary Fig. 3). Note that the **DAE-1o** alone is inert under 420 nm irradiation (Supplementary Fig. 3), demonstrating a possible participation of the triplet state during visible-light photochromism and the matched triplet energy levels between **DAE-1o** and **DT**. In contrast, **DAE-1o/PT** only exhibited slight photochromism (Supplementary Fig. 2b), indicating an unsatisfied triplet energy level matching between **PT** and **DAE-1o**. Notably, **DAE-1o/DT** hardly underwent photochromism in diluted solution (2×10^{-5} M, Supplementary Fig. 3), which presented an evident concentration dependence of photochromism in mixed system.

Supplementary Figure 2. Absorption spectra of **DAE-1o/DT** and **DAE-1o/PT** in deaerated toluene upon irradiation at 420 nm, respectively. $[\text{DAE-1o}] = [\text{DT}] = [\text{PT}] = 10^{-4}$ M.

Supplementary Figure 3. Absorption spectra of **DAE-1o** (1.0×10^{-4} M) in toluene upon irradiation at 313 nm and 420 nm, **DAE-1o/DT** (2.0×10^{-5} M/ 2.0×10^{-5} M) in deaerated toluene before and after irradiation at 420 nm.

Q2: p 7, line 2 from the bottom: The decrease in the lifetime of DT in aerated toluene indicates the oxygen quenching rate of 8×10^8 M⁻¹s⁻¹. Why is the oxygen quenching so inefficient?

A2: Thanks for your advice. We re-performed the whole set of transient absorption experiments more strictly, using the newly distilled toluene solvent and removing the oxygen by freeze-pump-thaw cycles method instead of previous bubbling nitrogen. A longer lifetime of 2973 ns in deaerated toluene was obtained (see revised Supplementary Fig. 5). Besides, we rechecked the lifetime of **DT** in aerated toluene and a shorter lifetime of 65 ns was obtained (see revised Supplementary Fig. 5). Based on these updated values, we calculated the oxygen quenching rate to be 4.9×10^9 M⁻¹s⁻¹, which is in accordance with that in previous articles (*J. Phys. Chem. A* **1999**, *103*, 5425).

We are so sorry for our carelessness in previous work, which might cause confusion to our readers. To prove the accuracy of our current results, we sequentially performed several sets of transient absorption experiments on **DT** triplet lifetime in different concentrations after aeration (see Fig. R6). The similar value of oxygen quenched triplet lifetime confirmed the validation of our updated results.

Supplementary Figure 5. Transient absorption decays of **DT** (1.0×10^{-4} M) at 620 nm in deaerated (black) and aerated (red) toluene, respectively.

Supplementary Figure 6. a) Transient absorption decays of DT (1.0×10^{-4} M) at 620 nm by adding different equivalents of DAE-1o in deaerated toluene; b) Stern-Volmer plot and linear fit for quenching of DT by DAE-1o.

Fig. R6 Transient absorption decays of DT in different concentration a) 5.0×10^{-5} M; b) 2.0×10^{-4} M at 620 nm in aerated toluene, respectively.

Q3: p 9, line 1-2 from the bottom: The rates have no meaning. These values are dependent on the illumination intensity.

A3: Thanks for your suggestion. We have removed the corresponding figure from Fig. 2.

Q4: p 11, line 3 from the bottom: The cyclization quantum yield of 0.23 in the presence of oxygen seems too large. It is recommended to show the transient absorption decay at 620 nm of DAE-o-DT in deaerated and aerated solutions, which may account for the oxygen effect on the cyclization quantum yield.

A4: Thanks for your suggestion. Fig. R7 shows the transient absorption decay at 620 nm (triplet absorption peak of DT) with a lifetime of 88 ns and 297 ns for DAE-o-DT in aerated

toluene and deaerated toluene, respectively. Considering the triplet life of **DT** alone is around 2973 ns, the oxygen quenching effect in our system is not that obvious probably due to the highly quenching efficiency from DT to DAE moiety. Similar oxygen quenching effect can be found in MLCT system, as the photocyclization quantum yield decreased from 0.44 to 0.22 in acetonitrile was reported previously in the paper (*J. Am. Chem. Soc.*, **2008**, *130*, 7286).

Fig. R7 The transient absorption decays of **DAE-o-DT** at 620 nm in a) deaerated; b) aerated toluene, respectively.

REVIEWERS' COMMENTS:

Reviewer #1 (Remarks to the Author):

I commend the authors for the improvement of the manuscript, expanding the set of experiments and even accepting that there were better interpretations with the new experiments. I believe that manuscript should be published. My only note is that ref 46 that has been added (Park et al) and is supposed to be about super-resolution imaging from photoswitching, is not relevant to the discussion at all. There are plenty of groups (Hell, Raymo, Enderlein, etc) who have published in that area, please pick a more appropriate citation.

Reviewer #2 (Remarks to the Author):

In this revised manuscript, authors successfully followed my comments to make supplementary experiments and to strengthen the ideas involved in this work. Additional synthesis for the longer spacer reference compound, static PLQY measurements, and rate constants evaluation etc. provided a very consistent and supporting evidences for the proposed TTET mechanism of DTE ring cyclization. I am now fully satisfied with this revised manuscript.

Reviewer #3 (Remarks to the Author):

The revised manuscript is worth publishing in Nat. Commun.

REVIEWERS' COMMENTS:

Reviewer #1 (Remarks to the Author):

I commend the authors for the improvement of the manuscript, expanding the set of experiments and even accepting that there were better interpretations with the new experiments. I believe that manuscript should be published. My only note is that ref 46 that has been added (Park et al) and is supposed to be about super-resolution imaging from photoswitching, is not relevant to the discussion at all. There are plenty of groups (Hell, Raymo, Enderlein, etc) who have published in that area, please pick a more appropriate citation.

A: Thank you very much for your comments. We have revised Ref. 46 according to your suggestion and a more appropriate reference is added.

“46. Roubinet, B. et al. Carboxylated Photoswitchable Diarylethenes for Biolabeling and Super - Resolution RESOLFT Microscopy. *Angew. Chem. Int. Ed.* **55**, 15429-15433 (2016).”

Reviewer #2 (Remarks to the Author):

In this revised manuscript, authors successfully followed my comments to make supplementary experiments and to strengthen the ideas involved in this work. Additional synthesis for the longer spacer reference compound, static PLQY measurements, and rate constants evaluation etc. provided a very consistent and supporting evidences for the proposed TTET mechanism of DTE ring cyclization. I am now fully satisfied with this revised manuscript.

A: Thank you very much for your comments, which really help us a lot to improve our manuscript. Even more, we consider this help us learn deep about our system and will provide a solid guidance to our future work. Thanks again!

Reviewer #3 (Remarks to the Author):

The revised manuscript is worth publishing in Nat. Commun.

A: Thanks a lot for your comments and suggestions!